# Structure and Chemical Composition of ca. 10-Million-Year-Old (Late Miocene of Western Amazon) and Present-Day Teeth of Related Species

**DOI:** 10.3390/biology11111636

**Published:** 2022-11-08

**Authors:** Caroline Pessoa-Lima, Jonas Tostes-Figueiredo, Natalia Macedo-Ribeiro, Annie Schmaltz Hsiou, Fellipe Pereira Muniz, José Augusto Maulin, Vinícius H. Franceschini-Santos, Frederico Barbosa de Sousa, Fernando Barbosa, Sergio Roberto Peres Line, Raquel Fernanda Gerlach, Max Cardoso Langer

**Affiliations:** 1Department of Basic and Oral Biology, FORP, University of Sao Paulo, Av. Café, Ribeirao Preto 14040904, SP, Brazil; 2Laboratory of Paleontology, Department of Biology, FFCLRP, University of São Paulo, Ribeirao Preto 14040900, SP, Brazil; 3Laboratory of Electron Microscopy, Department of Cell and Molecular Biology and Biopathogenic Agents, FMRP, University of Sao Paulo, Ribeirao Preto 14040900, SP, Brazil; 4Department of Biology, FFCLRP, University of São Paulo, Ribeirao Preto 14040900, SP, Brazil; 5Departament of Morfology, CCS, Universidade Federal da Paraiba, Cidade Universitaria, Joao Pessoa 58051900, PB, Brazil; 6Laboratory of Toxicology and Essentiality of Metals, FCFRP, University of Sao Paulo, Ribeirao Preto 14040900, SP, Brazil; 7Department of Biosciences, FOP, University of Campinas, Piracicaba 13414903, SP, Brazil

**Keywords:** teeth, enamel, microanalysis, *Purussaurus*, *Neoepiblema*, miocene, fossils, ICP-MS, MEV, microscopy

## Abstract

**Simple Summary:**

The dental enamel is the most mineralized tissue of vertebrates, and its preservation in fossil records is important to better understand the ancient life and environment on Earth. However, the association of morphological features with the mineral and organic information of this tissue is still poorly understood. This study aims to compare morphological features and chemical composition of dental enamel of extinct and extant species of alligators and rodents. Organic, mineral, and water content were obtained on ground sections of four teeth, resulting in a higher organic volume than previously expected (up to 49%). It is observed that both modern and fossil enamel exhibit the same major constituents: 36.7% calcium, 17.2% phosphorus, and 41% oxygen, characteristic of hydroxyapatite, the biomineral of vertebrates. Twenty-seven microelements were measured from superficial enamel. Zinc was the most abundant microelement, followed by lead, iron, magnesium, and aluminium. Semiprismatic enamel was observed in the alligator fossil. The fossilized enamel was in an excellent state for microscopic analyses. Results show that all major dental enamel’s physical, chemical, and morphological features are present both in extant and extinct fossil tooth enamel (>8.5 Ma) in both taxa.

**Abstract:**

Molecular information has been gathered from fossilized dental enamel, the best-preserved tissue of vertebrates. However, the association of morphological features with the possible mineral and organic information of this tissue is still poorly understood in the context of the emerging area of paleoproteomics. This study aims to compare the morphological features and chemical composition of dental enamel of extinct and extant terrestrial vertebrates of Crocodylia: *Purussaurus* sp. (extinct) and *Melanosuchus niger* (extant), and Rodentia: *Neoepiblema* sp. (extinct) and *Hydrochoerus hydrochaeris* (extant). To obtain structural and chemical data, superficial and internal enamel were analyzed by Scanning Electron Microscopy (SEM) and Energy Dispersive Spectroscopy (SEM-EDS). Organic, mineral, and water content were obtained using polarizing microscopy and microradiography on ground sections of four teeth, resulting in a higher organic volume than previously expected (up to 49%). It is observed that both modern and fossil tooth enamel exhibit the same major constituents: 36.7% Ca, 17.2% P, and 41% O, characteristic of hydroxyapatite. Additionally, 27 other elements were measured from superficial enamel by inductively coupled mass spectrometry (ICP-MS). Zinc was the most abundant microelement detected, followed by Pb, Fe, Mg, and Al. Morphological features observed include enamel rods in the rodent teeth, while incremental lines and semiprismatic enamel were observed in the alligator species. The fossil enamel was in an excellent state for microscopic analyses. Results show that all major dental enamel’s physical, chemical, and morphological features are present both in extant and extinct fossil tooth enamel (>8.5 Ma) in both taxa.

## 1. Introduction

Historically, advances in paleontology and the classification of extinct species have relied mainly on the comparative analysis of morphological aspects of fossilized material, especially in calcified structures such as bones and teeth. These analyses, however, may be constrained for some extinct species as the identification may be based on small fragments of fossilized material. The anatomical similarities between species may not directly result from shared phylogenetic history. Evolutionary convergence is often regarded as a relevant factor in morphological phylogenetics [1]. Paleogenomics and paleoproteomics have recently appeared as essential tools that may assist in classifying extinct species.

Among calcified tissues, tooth enamel has the highest mineral content [2], corresponding to higher density, smaller pores (nanopores) [3], and a unique set of proteins, whose recovery from superficial enamel etches started only in 2006 [4]. This innovation was based on the “enamel biopsy”, a superficial enamel etching used to obtain ultraclean samples. Such samples enabled the comparison of the exposure of children to lead, a neurotoxin [5,6,7]. The superficial enamel sample had two main advantages: teeth did not need to be ground/destroyed to be studied anymore, and the spatial information on the number of chemical elements on the outer enamel (where many elements with high affinity for hydroxyapatite accumulate) would not be lost. The need to develop ultraclean techniques to be able to obtain reliable results on the superficial etchings was a key factor that later resulted in excellent results from the mass spectrometry analysis of biopsy samples [3,8,9].

Nowadays, enamel proteins recovered from ancient species are considered particularly useful for comparing divergent species [10,11,12]. As a “pre-fossilized” extracellular matrix, dental enamel is superior to other calcified tissues in retaining the features of living specimens [13,14]. The inorganic composition of fossil teeth makes it possible to understand aspects such as diet, species’ migration patterns, and paleoenvironmental interpretation [15,16]. Enamel peptides have been recovered from fossil enamel, indicating that the unique properties of this tissue act as a protective barrier preventing protein degradation [17].

Many gaps in knowledge exist, as expected in a very new field of research. To date, information on the association of morphological features of dental enamel with the composition (quantity and diversity) of proteins in forming enamel is still poorly understood. In recent years, this information has been obtained primarily from mouse models [18,19,20,21]. Little is known about other species, since the protein information is gained while enamel is forming, and this usually happens in early life. Such information would be very beneficial for the analyses of peptides in different species. Since proteins are the molecules that form the scaffold that organizes crystal growth, it is intuitive to assume that different amounts (quantitative changes) of proteins will determine different phenotypes of dental enamel, and not only the mutations accumulated over time (qualitative changes). So, it may be that both different amounts of the different enamel proteins and different enamel protein sequences will have an influence on the physical and chemical properties of the enamel. This seems particularly important in light of the tooth enamel phenotype observed in transgenic and enamel proteins of knock-out mice, which offers the chance to look into experiments that nature might have tested over large scales of time.

Therefore, it seems that it might be useful to try to associate morphological features with the possible mineral and organic information of this tissue. This information contributes to better interpreting molecular information that is now being gathered in several excellent research centers, where the new area of paleoproteomics is being developed.

In this study, we compare the morphological features and the chemical composition of dental enamel of extinct and extant terrestrial vertebrates belonging to two groups. Crocodylia was represented by *Purussaurus* sp. (fossil) and *Melanosuchus niger* (extant), while Rodentia was represented by *Neoepiblema* sp. (fossil) and *Hydrochoerus hydrochaeris* (extant). We combined several techniques adapted to study developing enamel [22,23,24], and we used polarized light microscopy and birefringence to determine organic content [25]. The mineral composition was studied with ICP-MS, as opposed to other chemical analyses that are less sensitive [26]. This study shows for the first time the mineral, organic, and water contents of the enamel of extinct and extant species from the same taxonomic families, and some of the species show high organic content.

## 2. Materials and Methods

### 2.1. Institutional Abbreviations

UFAC—Universidade Federal do Acre, Rio Branco, AC, Brazil.

LIRP—Laboratorio de Ictiologia de Ribeirao Preto, FFCLRP, University of Sao Paulo, Ribeirao Preto, SP, Brazil.

### 2.2. Fossil Specimens and Provenance

Both fossil specimens used in this study came from the Acre Basin, a retroarc foreland basin related to the Andean orogenesis [27]. It is the westernmost of a series of interior sag/fracture basins along the Solimões/Amazonas rivers, at the westernmost portion of the Brazilian Amazon, neighboring Peru and Bolivia [28,29,30]. They were collected in the sedimentary deposits of the Late Miocene Solimões Formation, which is mostly exposed along riverbanks in the states of Acre and Amazonas. Such sediments were deposited mainly under fluvio-lacustrine conditions, both in river channels and in floodplains/lakes [31,32]. A diverse vertebrate fauna has been reported in the Solimões Formation, including cartilaginous and bony fishes, frogs, turtles, birds, crocodylians, lizards, snakes, and mammals (e.g., [33,34,35,36,37]).

UFAC-7226 (Figure 1)—the first sample corresponds to a *Purussaurus* sp. tooth. It was collected in the locality known as “Niterói”, which is located on the right embankments of the Acre River, Senador Guiomar municipality (UTM 19L 629983 E/8879539 S, datum WGS84). The fossil comes from the same bone-beds U-Pb dated by Bissaro-Júnior et al. [30], from detrital zircon, with a maximal weighted-mean age of 8.5 ± 0.5 Ma (Tortonian Stage, Late Miocene). *Purussaurus* was one of the largest crocodiles, reaching more than 10 m in length [38], with records in the Miocene of Brazil, Colombia, Peru, Venezuela, and Panama. *Purussaurus* is regarded as a semiaquatic top predator, placed in recent phylogenetic studies within Alligatoridae and Caimaninae, forming, with other taxa, the sister clade to Jacarea [39].

UFAC-7227 —the other fossil sample analyzed consists of a *Neoepiblema* sp. partial lower incisor. It was collected in the locality known as “Talismã”, which is located on the right embankments upstream of the Purus River, State of Amazonas, between the Manuel Urbano and the Iaco River mouth (UTM 19L 510475 E/9029741 S, datum WGS84). The fossil comes from the same bone-beds U-Pb dated by Bissaro-Júnior et al. [30], from detrital zircon, with a maximal weighted-mean age of 10.89 ± 0.13 Ma (Tortonian Stage, Late Miocene). *Neoepiblema* is a large-sized, caviomorph rodent, with records in the Miocene of Brazil, Peru, Venezuela, and Argentina. Along with other neoepiblemids, it is more closely related to chinchillids than to other caviomorphs [40].

### 2.3. Modern Specimens

Teeth of two extant species, *M. niger* and *H. hydrochaeris*, were analyzed to compare the structure and chemical composition of enamel of the modern and fossil samples. *M. niger* is a South-American alligator with geographical distribution in the north of the continent. *H. hydrochaeris* is the largest living rodent and has a widespread geographical occurrence in South America. The specimens’ teeth were obtained from LIRP’s zoological collection. The *M. niger* specimen was from "Reserva Mamirauá", Amazon State, and the *H. hydrochaeris* specimen was from Ribeirao Preto, São Paulo State.

### 2.4. SEM-EDS Analysis

Analyses by Scanning Electron Microscopy (SEM) using conventional and Back-scattered-electron mode (BSE) were performed on a scanning electron microscope (Jeol JSM–5600LV, Tokyo, Japan) [22,41]. EDS signals were captured using a standard setup on BSE mode. The EDS signals were obtained using the following parameters: Samples were positioned at a 10 mm distance from the emitter, with 15 to 25 kV of emission. Spot size ranged from 69 to 80 nm. Measurements were made on the surface of the dental enamel. For the SEM-EDS analysis, internal multielement standards were used for calibration according to the supplier (Oxford Instruments, Scotts Valley, CA, USA). Based on those standards, the maps of the most abundant elements and quantifications were made [42]. The *Purussaurus* tooth have three different superficial colors. Thus, three regions on the tooth surface were chosen for analysis: (1) a black area (visible in lines), (2) a brown area (most abundant), and (3) a yellow area. The images were taken in both the conventional and Back-scattered-electron (BSE) mode in different magnifications. Multielement maps were produced automatically by the Aztec program (Oxford Instruments, Concord, MA, USA) based on the intensity of the signals of the different elements. The detector used was the Silicon Drift Detector (SDD, X-MaxN 150 mm${}^{2}$ detector).

### 2.5. Preparation of Samples for Light and Scanning Electron Microscopy

Pieces of teeth were embedded in acrylic resin (JET, Campo Limpo Paulista, SP, Brazil) and cut with a diamond disc on a calcified tissue cutting machine (Elsaw, Elquip, São Carlos, SP, Brazil). Sections were carefully sanded with a gradual decrease in the sandpaper granulation, from 600 to 4000, using water. The width of the sections was circa 100 μm for bright field and polarized light microscopy. For SEM, thick sections were used, which were etched with 10% acetic acid (*Purussaurus* sp. enamel) for 15 s or with 37% phosphoric acid for 30 s (enamel from the other 3 species). Immediately after acid etching, the teeth were copiously washed with running water for 10 min to remove traces of the acid. Samples were then dehydrated overnight and covered with a gold pellicle. The SEM photographs were taken at the same microscope described above.

### 2.6. Light Microscopy

Tooth sections were photographed on an Axiovert 2.1 microscope (Zeiss, Karlstadt, Germany) with and without phase contrast filters, and on an Axioscope 40 (Zeiss, Karlstadt, Germany) using crossed polarizing filters and the λ/1 red filter.

### 2.7. Quantification of Major Enamel Biochemical Components

Major enamel biochemical components (mineral, organic, and adsorbed water in volume and weight percentages) were measured in each sample (non-demineralized ground sections with thickness in the range of 100–400 μm), at five discrete regions of interest (ROI; 12 μm × 12 μm) along a longitudinal line running from the enamel surface to the enamel dentine junction, as described previously in detail [43,44]. Unlike thermogravimetric analysis [45], the methods used here did not require the destruction of the samples. Basic physical parameters related to the mineral, organic, and water contents (mineral unit-cell composition, mineral X-ray linear attenuation coefficient, mineral density, and refractive indexes of all major biochemical components), required for quantification, were derived from human enamel. Transverse microradiography [46], the gold-standard technique for quantification of dental enamel mineral volume [47], was used to quantify mineral content. The following assumptions were considered for the enamel mineral composition: unit cell composition of Ca${}_{8.856}$Mg${}_{0.088}$Na0.292K${}_{0.010}$(PO4)${}_{5.312}$(HPO${}_{4}$)${}_{0.280}$(CO${}_{3}$)${}_{0.407}$(OH)${}_{0.702}$Cl${}_{0.078}$(CO${}_{3}$)${}_{0.050}$) and density of 2.99 g/cm${}^{3}$[48], with 37% calcium weight and 18% phosphorus weight (Ca/P ratio of 2.06). Digital 2D microradiographic images (enamel ground sections and aluminium step-wedge with 17 aluminium foils, with thickness ranging from 20 to 340 μm) were obtained from microtomographic equipment (Skyscan 1172, Bruker, Belgium) operated at 60 kV, emitting peak X-ray energy of 10 kV [43]. After mineral volume quantification, birefringence measurements (mean of five measurements) using underwater immersion (and after immersion in water for 5 days prior to the analysis) were obtained using polarizing microscopy (Axioskop 40, Carl Zeiss, Germany) equipped with a 0–5 orders Berek compensator, and then organic and water volumes were quantified from the mathematical interpretation of birefringence [25,44]. The assumed refractive indexes of organic matter and water were 1.56 and 1.00, respectively. The sum of all measured volumes resulted in 100% enamel volume at each ROI. Volume percentages were converted to weight percentages using the following densities: 2.99 g/cm${}^{3}$ (mineral), 1.45 g/cm${}^{3}$ (organic; glutamic acid), and 1.0 g/cm${}^{3}$ (water).

### 2.8. Superficial Enamel Sampling with Acid

The outer enamel of all 4 teeth was etched twice. This method is ideal to obtain superficial etches for microelement determination and the study of environmental contaminants in children in vivo [5,49]. The method was used by Brudevold et al. [50], with the biopsy depth and other parameters studied in detail [51]. This method was then adapted to obtain enamel peptides by Porto et al. [2,52], having been also used by Stewart et al. [8,9] and Nogueira et al. [3]. In the *Purussaurus* sp. tooth, 3 different regions were selected for sampling due to the color differences, as described above, and as seen in Figure 1. This procedure was carried out inside a Class 100 hood, using only ultraclean solutions diluted just before use. The biopsy was analyzed in order to perform the determination of 27 chemical elements by Inductively Coupled Plasma-Mass Spectrometry (ICP-MS). In this study, lead-free adhesive tape (Magic Tape, 810 Scotch^®^, 3M, Sumare, SP, Brazil) with a circular perforation (diameter = 2 mm) was firmly pressed onto the surface of the tooth, delimiting the microbiopsy site (the tape with the circular perforations can be observed in Figure 1A). The sampling site was etched according to the following procedure. The enamel surface was rinsed with water (MilliQ) and etched with 20 L 10% (vol/vol) HCl for approximately 20 s. The microbiopsy solution was then transferred to an ultraclean centrifuge tube (1.5 mL) (Axygen Scientific, Inc., Union City, NJ, USA). The samples were closed with parafilm and sent for microelement determination by ICP-MS (NexIon 5000, Perkin Elmer) in the Laboratory of Toxicology and Essentiality of Metals (University of São Paulo, FCFRP, Ribeirão Preto, SP, Brazil), without opening the tubes before the analyses. The second etch was made in the same way, but the acid was then used to obtain enamel peptides, as described in a separate manuscript.

### 2.9. Chemical Analysis by ICP-MS

Several macro- and microelements were determined on the superficial enamel samples by Inductively Coupled Plasma-Mass Spectrometry (ICP-MS) in the Laboratory of Metals Toxicology, University of Sao Paulo in Ribeirao Preto (Brazil). The quantification limit (LOQ) for the different elements in the study is presented in Appendix A. Calcium concentrations determined by SEM-EDS in each sample (Table 1) were used to calculate the mass of dental enamel. Based on this, the amount of each element was expressed in ppm (μg/g) of dental enamel, as a more universal concentration.

## 3. Results

Here, we compare 10-Million-year-old fossils and modern enamel from related species of two different vertebrate taxa. For clarity, we first describe the macro- and microscopic features of the teeth and enamel, while the chemical composition is described later.

### 3.1. Morphological Description

Figure 1 shows images made from the fossil alligator tooth used to obtain superficial enamel acid etch. The tooth measured ≈7 cm in length and ≈4 cm in width (Figure 1A). Figure 1B shows a larger view of the white rectangle that can be observed in Figure 1A. Small cracks on the surface form a fairly regular rectangular lattice. The area etched with acid is the small circle, which has a ca. 2 mm diameter. This image shows that the etching only slightly modified the superficial aspect of the enamel and that such changes can only be seen under magnification. This image also shows the variation in color seen in the enamel, as well as the cracks at right angles. Some gray “squares” can be seen in the lower part of the figure, where the enamel was unintentionally removed together with the tape. The gray observed in the picture is the underlying dentine. Another observation here is that the enamel in this fossil sample is poorly attached to the dentine. The light strength with which the glue on the tape was attached to the enamel was sufficient to detach the enamel from the dentine in this area with many cracks. The many details in shape and color seen in the enamel greatly contrast with the gray appearance of the dentine and are a clear indication that enamel does act as a reflective surface for light, and this indicates the very high crystallinity of this calcified tissue. The same was observed in all other specimens, both fossil and modern. The underlying dentine, on the other hand, displays a gray appearance that is compatible with the incorporation of other ions into its structure, and also a greater ability to absorb light, the opposite of the reflection of light seen in the enamel. Figure 1C–E demonstrates that the acid etching extraction technique causes minimal damage to the fossil surface. The superficial aspect of Figure 1E shows a rough surface on the etched area.

Figure 2 shows images made from the extant alligator tooth. In Figure 2A, the whole tooth is shown, and the dental enamel covers the tooth crown, which is seen to the right side of the arrow and displays a yellowish appearance, except for the area etched with acid, amplified in Figure 2B. The enamel of this species has a porous superficial appearance where the acid was applied.

In comparison wish the alligator fossil enamel, the modern enamel does not show many right-angle cracks. We believe this difference can be the result of weathering before the burial of the fossil or lithostatic compression.

Figure 3 shows a fragment of the fossilized rodent tooth. The superficial enamel is shown in Figure 3A. The enamel has a brown aspect and has some white deposits. On the left side of this image, the underlying dentine is apparent. In Figure 3B, the etched enamel is seen as a circle, showing minimal damage to the specimen (in B, the bar is 1 mm, and the circular area has a diameter of 2 mm). In Figure 3E, the prismatic enamel is seen in the etched enamel.

In sharp contrast to the *Purussaurus* sp. enamel, no cracks were observed. It is possible that alligator’s fossils have being exposed to greater taphonomic effects than the rodent’s fossils, or the fossils have reacted differently to similar conditions.

Electron Microscope using Back-scattered-electrons (BSE) mode (**C**–**E**). Rectangles are always amplified in the next figure of the series. (**A**). Picture taken without a magnifying lens; the arrow indicates the limit between the crown (to the right) and the root (to the left). The crown exhibits a yellow appearance on its surface; Bar= 1cm. (**B**). Details of the rectangle are depicted in (**A**); the area etched with the acid is shown as a circle, but some acid has leached to the sides, as seen from the brighter areas on the enamel. Bar = 1 mm. (**C**). Amplification of the rectangle shown in (**B**). Bar = 500 μm. (**D**). Amplification of the rectangle shown in (**C**). Bar = 100 μm. (**E**). Amplification of the rectangle shown in (**D**). The high porosity of the etched enamel is clearly observed. No enamel prisms can be observed. Bar = 50 μm. 

Figure 4 shows the tooth enamel of the extant rodent. The superficial nature of the enamel biopsy can be observed in Figure 4A,C. Circular transparent spots with a diameter of ≈200 μm can be observed. Figure 4C is an amplification of the rectangle shown in B. The circular area was etched with acid. The circular spots are superficial, having disappeared in the etched area, and have a composition with less dense elements (suggesting an organic composition) than the Ca and P that compose the hydroxyapatite of the enamel. Figure 4D is an amplification of the rectangle is shown in Figure 4C, and Figure 4E is an amplification of the rectangle shown in Figure 4D. Prismatic enamel is seen on the left (asterisk), in the etched area.

### 3.2. Microstructure (SEM Analysis)

Figure 5 exhibits SEM (conventional method with gold coat) pictures of cross sections of the the enamel of the four species. The enamel of the fossilized alligator (Figure 5A–C) is formed by rod-like structures that are comparable in shape and width to the prismatic structures of mammalian enamel. These structures run roughly perpendicular to the enamel surface and dentine-enamel junction (DEJ), having a width of approximately 5 μm. In extant alligators (Figure 5D–G), there is a typical aprismatic structure, showing growth lines that are likely to be equivalent to the mammalian Retzius lines. These lines were localized near the DEJ and likely represent periods of physiological stress during enamel formation. The enamel of the fossilized rodent (Figure 5H–K) is formed by prisms that are roughly parallel to each other and perpendicular to the (DEJ) and enamel surface. The DEJ exhibits an irregular surface. The enamel of the extant rodent shows two distinct patterns of the arrangement of the prisms. The prisms run nearly parallel to the DEJ and extend approximately 25–30 μm near the DEJ. A thin layer of parallel prisms is also observed near the enamel surface. In the inner enamel, a different pattern is observed, with a large band (≈80 μm) of prisms exhibiting intense decussation.

Appendix A shows photomicrographs of ground sections (≈100–400 μm) of the four teeth used in this study. The images show that enamel (e) is translucent in all specimens, while dentine (d) is dark/opaque in all specimens. We show an exception to this, where the mineral has been lost, possibly due to acids from the environment (Appendix A). The white arrows in panels A and C show marked apposition lines of the dentine. In D, black arrows in the enamel show appositional lines. In F, the arrow indicates a very pronounced line; the inner enamel is less mineralized, and the outer enamel appears more mineralized. In G, the enamel of the extant rodent is shown; the arrow indicates a less mineralized area in the middle of the enamel throughout the enamel. In H, a larger magnification of the black square is shown under polarized light. Due to the thickness of this section, the prismatic aspect of this enamel is difficult to recognize, but in the outer enamel (asterisk) diagonal lines are enamel rods.

### 3.3. Determination of Organic, Mineral, and Water Content of Dental Enamel

In this study, major enamel biochemical components (mineral, organic, and adsorbed water in volume and weight percentages) were measured at five discrete regions of interest (ROI; 12 mm × 12 mm) along a longitudinal line running from the enamel surface to the DEJ in the enamel of each tooth.

Figure 6 and Figure 7 show the images made at each step of the initial analysis of the fossil tooth ground sections from alligator and rodent fossils under polarizing light. The sections were submerged in water during sample preparation. Without an interference filter, dental enamel shows high interference colors (Figure 6A,B and Figure 7A,B), indicating a relatively high ground section thickness. With the Red I filter, the additional interference color is shown at the −45° (45° counterclockwise) diagonal position (Figure 6C), and the subtraction interference color is shown at the +45° (45° clockwise) diagonal position (Figure 6D), indicating negative birefringence, based on the Michel-Levy interference color chart [53].

The step-by-step analyses of the modern samples are not shown here, but figures of polarized light microscopy and microradiography are shown for each sample as Appendix A. The enamel of all samples exhibited negative birefringence.

Table 2 summarizes the mineral, organic, and water volume (vol%) and weight (wt%) obtained based on the Equation used for human enamel, which also exhibits negative birefringence. These values refer to five ROI (regions of interest) selected in the enamel layer, taking into account that the enamel thickness (distance from enamel surface to the DEJ varied among the samples. The enamel thickness of the fossilized alligator is 115 μm; the enamel thickness of the extant alligator is 190 μm. The fossilized rodent has an enamel thickness of 205 μm, while the enamel thickness of the extant rodent is 145 μm. The density of the organic component was estimated based on the density of glutamic acid (1.5 g/cm${}^{3}$). The "water" component corresponds to the adsorbed water located in the enamel pores, outside the mineral crystallites.

The mineral vol (%) found in the enamel of the fossilized alligator varied from 83.4 to 89.2, corresponding to 92 to 95.3 wt%. The organic matter vol (%) was 8.1 in the outer enamel, corresponding to 4.5 wt (%), with an increase in the middle part of the enamel (10.3% at 55 μm, corresponding to 5.7% by weight) and a decrease in the organic content in the deeper enamel (5.8 vol%; and 3.1 wt%). The water content did not vary more than 0.2 from the outer to the inner ROI, starting at 6.6 vol (%) and decreasing to 6.4% in the inner enamel, corresponding to 2.4 and 2.3 (wt%), respectively.

In the sample of the modern alligator enamel, mean mineral content was 59.4 (vol%) and 75.2 (wt%). The values for organic volume ranged from 33.9 to 49% (vol), corresponding to 20.98 to 33.51% (wt). The water content varied from 1.17 to 5.18% (volume), corresponding to a mean water wt (%) of 1.4. The biggest difference in mineral and organic matter was found between the surface of the enamel and the second ROI, with less mineral and higher organic content in the surface, as compared with the 2nd to 5th ROIs.

The measurements of the fossilized rodent enamel showed a mean mineral content of 52.3 (vol%) and 70 (wt%) and an organic content of 39.2 (vol%) and 26.3 (wt%). The largest difference between different depths was found in the outer enamel, with the first ROI indicating an 8 (wt%) difference compared with the 2nd ROI, which was more homogeneous with the deeper layers analyzed. The outer layer displayed less mineral and a higher organic content. The water content varied from 8.04 to 10.13% of total enamel volume, and 4.1 (wt%).

The extant rodent enamel showed a very homogeneous mineral content, ranging from 68.1 to 71.6 vol% (82.9–85.3 wt%), and an organic content of 18.5 ± 1.2 (vol%), corresponding to 11.2 ± 0.8 (wt%). The water content exhibited little variation between the ROIs, with a mean vol% of 11.7 (±0.4), corresponding to 4.8 ± 0.2 (wt%).

Appendix A show graphs that visually represent the variation of biochemical content along the enamel depth.

### 3.4. Inorganic Composition of the Superficial Enamel

#### 3.4.1. Energy-Dispersive Spectroscopy (EDS)

Relative quantification of the major chemical elements by SEM-EDS resulted in a ratio of 2.0–2.2 between Ca and P for both modern and fossil enamel (Table 1), characteristic of hydroxyapatite. These results corroborate the previous assumption of using the unit cell composition proposed by Elliot [48] for the determination of organic, mineral, and water content of dental enamel.

#### 3.4.2. Inductively Coupled Plasma-Mass Spectrometry (ICP-MS)

The ICP-MS analysis results identified 27 elements from superficial enamel samples. Calcium values were used to determine the amount of enamel obtained in each sample. Thus, the results of 27 microelements found in each sample are expressed as μg/g of dental enamel (Figure 8). The most abundant element was Zn (4–23%), followed by Pb (0.05–3.7%), Fe (0.5–1.7%), Mg (0.2–1.5%), Al (0.06–1.4%), a series of elements with values near 0.5% (K, Cd, Cr, Mn, and Co), and other still less abundant elements (Be, V, Ni, Cu, As, Se, Rb, Ag, Ba, Tl, Bi, Pd, Th, La, Ce, Sm, and U) near or lower than 0.01%. Not all elements were detected in all samples. The results are summarized in Figure 8 and Figure 9, and all values are presented in Appendix A.

An increase in the concentration of Pb, Co, Cd, Ce, Th, As, Cu, Bi, Ag, Tl, and U (U234, U235, and U238) is seen in the fossilized enamel, with different magnitudes of change (Figure 8). No expressive difference between fossil and modern samples was detected for the other microelements.

Figure 9 summarizes the results of this study regarding the enamel inorganic element composition. The macroelements (Ca, O, P, and C), determined by SEM-EDS are shown in the pie chart. The microelements, represented by the gray slice of the pie, cannot be estimated by SEM-EDS but might be <2%. A tentative relative amount (%) is drawn based on the ICP-MS results for the microelements. The total of microelements recovered from ICP-MS analysis was transformed to 100%, and based on this, the right-side chart was constructed for microelement distribution. Thus, each element appears with a relative proportion. At first glance, the most abundant microelement is Zn, followed by Pb or Fe, depending on the sample. Note that Fe (shown in orange) has a similar absolute percentage (Appendix A) in extant alligator (1.7%) and fossil rodent (1.5%), but this similar relative amount is not evident in Figure 9, since no Zn was measured in extant alligator.

Since the enamel of extinct alligator was etched in three different areas (1. Darkest region of enamel; 2. The predominant brown color of enamel; and 3. Yellow region of enamel), these areas are mapped in the upper-left part of Figure 9, so that the results from each area can be separately shown in the graph.

## 4. Discussion

This study highlights several important morphological and chemical aspects of modern and fossil enamel. Since several points regarding the recovery of ancient organic molecules in fossils have been questioned in the last decade, this broader investigation was undertaken with specialists from several areas of the dental enamel research, and a smaller number of samples, to be able to make more precise determinations.

This study brings to light some points that can attract more attention in the future, so that higher quality information can be gained from fossils, both in the interest of better understanding early life in ancient environments, and also, possibly, the metabolic consequences of the evolution of cells and proteins and how they were formed, folded, and worked in environments with different metal concentrations. The discrepancy between the availability of chemical elements (mainly metals) in modern and ancient environments is what makes some chemical elements nowadays toxic to organisms, such as lead, which has a high affinity for hydroxyapatite due to its chemical similarities with calcium. Skeletons were used to prove the anthropogenic contamination of the Earth with lead based on the capacity of bone to harbor lead in its mineral for long times such as decades or centuries [54].

This study started based on the need to have a more solid knowledge of the similarities and differences between modern and fossil samples due to our interest in recovering ancient enamel-specific peptides. Although it is well-known in paleontology that teeth are the best-preserved fossils, and sometimes only enamel is recovered in some fossils, the comparison of modern and ancient enamel has been performed only by a few groups. To our knowledge, so far such studies were carried out to observe chemical aspects, with a particular interest in the diagenetic processes [55,56,57] and isotopic analysis [16,58,59] and, as we did before, aiming at recovering protein material from fossils [3,11,12]. Another set of studies did show morphological features of dental enamel [13,14], which had already demonstrated that microscopic aspects were well-preserved. However, those studies did not establish any association between the well-defined morphology of prisms and other structures, with the possibility that such preserved structure might also indicate that fewer (bio)chemical alterations had taken place over time. Thus, this study started from the observation that similar physical aspects were observed (even by the naked eye) in the enamel of modern and fossilized species of correlated taxa: this was a consequence of the high crystallinity and high density of the enamel that seemed to have been preserved.

To the best of our knowledge, this study reports, for the first time, spatially resolved major enamel biochemical component volumes at discrete histological layers of dental enamel. The sum of the organic and water volumes constitutes the pore volume, which, summed with mineral volume, yields the total enamel volume. Corroborating the enamel mineral theoretical composition used for microradiography, with a Ca/P ratio of 2.06 [48], experimental SEM-EDS data indicated Ca/P ratios in the range of 2.0 to 2.2 (Table 2). Two types of water have been reported in dental enamel, lattice water (inside the mineral crystallites) and adsorbed water (located in the pores, outside the mineral crystallites) [60]. The water content measured here represents the adsorbed water, the main pathway for the diffusion of materials in dental enamel. Similar to sound human dental enamel [43,44,46], all samples presented negative birefringence under water immersion, typical of dental enamel.

The intensity of negative birefringence in (mature) enamel is directly proportional to the solid volume (mineral and organic), so negative birefringence is explained by an increased organic volume in sites with low mineral volume [25,44]. In humans, as other mammals who use teeth intensively for food breakdown in preparation for initial digestion by salivary enzymes in the oral cavity, average mature mid-crown dental enamel presents 93% mineral volume (based on a mineral density of 2.99 g/cm${}^{3}$), 5.5% water volume, and 1.5% organic volume [25,46]. From the occlusal to cervical regions of the tooth crown and from outer to inner enamel, following a decreasing gradient of mastication-related mechanical load, mineral content decreases [61], accompanied by an increase in both water and organic contents [62]. Possibly, variations in major enamel component volumes described might reflect variations in the use of teeth for mastication, the occurrence or not of continuous amelogenesis, and the rate of tooth replacement. Particularly, by contrasting the fossil with the extant alligator, the higher mineral content of the *Purussaurus* sp. (fossil) enamel might be explained by the more intense use of teeth for food breakdown, with a wider range of lower jaw movements, though other specimens (extinct and extant crocodiles) need to be analyzed. Overall, our data suggest that all samples provide a relatively large amount of organic matter and would be suitable sources of enamel peptides. There are, though, several points of uncertainty that can lead to much better results with dental enamel in the future. In particular, the determination of mineral composition, organic and water volumes, and weight percentages, as described in this study, is based on human enamel density, and this is clearly an assumption that is unlikely to reflect the reality in the enamel of other species, and the dental enamel density in fossils also needs to be determined. However, this can be achieved.

Differences in the elements found in modern and fossilized specimens seem to reflect differences in the permineralization of the fossil. The higher mineral content of the alligator fossil enamel, as compared with the rodent fossil enamel, may reflect that they came from different localities, as well as the local differences in the mineral content of the sediments from which the fossils were recovered.

Furthermore, the determination of organic material will certainly benefit from the development of other forms of microscopic techniques to help the determination of organic composition, such as the slow decalcification of samples after the use of alternative fixatives, as already used in the 1960s for microenzimology and successfully applied to recover enzyme function in mineralizing teeth [24,63]. Because the dental enamel is so calcified, techniques normally used for other tissues are difficult to apply to this tissue, but the determination of proteins will be possible with some developments. Interestingly, in the 1960s, alternative fixatives were developed to avoid the use of aldehyde-based fixatives that destroy enzyme activity. This fixative is also incompatible with mass spectrometry determination of peptides and proteins since they cross-link different parts of proteins covalently. So, certainly, there is much to be learned from fossil enamel using alternative fixative methods and mass spectrometry.

The discussion of the organic, inorganic, and water content of the different depths of human and mouse enamel (the ones that have been more studied so far) would be long and outside of the scope of this article. However, there are already several independent lines of evidence (based on different techniques) that suggest that 96 wt% of the mineral in dental enamel might only be found in the very superficial human enamel and possibly also in other mm-thick enamels of mammals. The very thick enamel of mammals, as far as is known, has a very long mineralization process that results in the high mineral content of the outer surface [64]. There is also long-standing evidence that protein remains in the enamel in higher amounts, and careful microscopic technique has been able to show remaining scaffolds of proteins (called “tuft” protein in the past), as discussed by Robinson and Hudson [65].

According to Smith et al. [19], in a study on the comparative proportion of mineral and volatiles in the developing enamel of normal and genetically modified mice, “volatiles were also found in amounts that were often higher than expected, especially in more mature enamel”. This study also showed some other interesting aspects that must be kept in mind in the future when techniques “adapted” to studying enamel will (hopefully) be used for many more groups. One such aspect is, for instance, the idea that large amounts of protein will not hinder enamel maturation, as observed in some genetically modified mice. As stated by Smith et al. [19], “the results of this study suggest that maturational growth of enamel crystals can occur in the presence of relatively large amounts of proteins and/or their fragments. The crown problems […] have nothing to do with protein or mineral but they occur because amelogenesis is shut down early and the enamel organ cells transform into a dysplastic epithelium that secretes a thin calcified material that is not enamel”. Appendix A shows a superficial layer in the extant alligator enamel that appears to be enamel but might be an indication that “amelogenesis was shut down early” in this specimen, which is a modern tooth that showed higher content of organic material.

It is the fact that nature has time and a countless number of possibilities that makes the diversity of phenotypes so great. The study of ancient species brings to light unknown structural aspects that resulted from genetic variation. Since genetically modified mouse enamel has been studied in detail for most enamel proteins, the correlation of some findings from such models with structural aspects of past species’ enamel might help us better understand past organisms. This is the case in transgenes that overexpressed ameloblastin, an enamel protein that is secreted in the secretion phase of amelogenesis and is part of the calcium-binding phosphoprotein (SCPP) family that evolved from a common ancestral gene [66,67]. Higher levels of expression of ameloblastin in a background lacking amelogenin resulted in short and randomly oriented apatite crystals, and the enamel resembled that of the reptile *iguana* [68].

In addition to many other implications, the variety and different proportions of enamel proteins have an impact on the research looking for ancient peptides and proteins. Structural and ultrastructural features might provide indications for the set of enamel proteins that can be found for paleoproteomics research. The ancient proteins might be preserved longer than we expect, but we might not be able to recover them so efficiently without knowing their sequences and post-translation modifications, as well as how they interact with each other and with minerals.

Bartlett et al. [21] showed that the knockout of Amelx and Mmp20 in mice resulted in the presence of “fan-like” crystal arrangements in the deeper enamel (20 μm of enamel, close to the DEJ), while the middle and outer enamel of such animals show a dysplastic and more mineralized layer [21]. Interestingly, the enamel of the extant alligator exhibited mineral structures that resemble fans in form (Figure 5F–G), with the same orientation (the larger border of the fan towards the outside of the enamel). Additoinal studies of the modern alligators’ enamel can provide more information on the enamel proteome, the detailed crystal arrangement, and, possibly, the relationship between both.

The ICP-MS data must be viewed with caution, since only a single measurement was made. Nonetheless, though the exploratory results of this study, we consider these data important for their contribution to a wider view of some aspects of the enamel in modern and fossil samples, and they may help formute new hypotheses and the need to avoid some of the mentioned problems of this study in the future. The lack of Zn in one of the samples is one of the problems. This is an error that cannot be fixed, since only one sample was taken for measurement, and might result from the high levels of microelements found in blank samples and the exclusion of many data based on this. Zinc is a very important mineral for metabolism, is present in many proteins (particularly in enzymes), and is always found in dental enamel. The results of this study are preliminary and need to be repeated in a larger set of samples, with samples collected at least in triplicates.

The time scale seems to be important to explain the presence of lead in higher concentrations on the outer enamel of fossils, in addition to other factors such as the acidity of the soil and the fossilization process. This accumulation on the outer enamel over millions of years is different from the low levels of lead found in the enamel of most humans or animals (that do not live in contaminated areas). It is clear that superficial enamel can lose minerals as it can also gain minerals (a net loss of Ca and P being the cause of caries). During these pH changes, lead can be incorporated into the enamel, even to deeper enamel [69]. However, this does not change the fact that lead on the surface of modern human enamel reflects the contamination of the environment [5,7,70,71,72,73], and the fact that in pre-industrial times the composition of superficial enamel is expected to have less lead, as lower lead levels were found in skeletons of prehistorical humans [54]. Therefore, the fact that lead and other minerals show a gradient in surface enamel (F, Cu, and other metals also show this gradient [74]) points out the need to carefully analyze superficial enamel, taking into account time scale and how many micrometers from the surface the ions were determined.

## 5. Conclusions

Specimens analyzed in this study showed high crystallinity and high density in both modern and fossilized enamel. Few structural changes were detected relative to the age of the sample. The polarizing microscopy indicated the birefringence property for the modern and fossilized samples (negative birefringence). The microradiography determination of major enamel biochemical components showed unexpectedly high organic weight (%) in two specimens (23.72% in extant allitagor and 26.30% in fossilized rodent). Relative quantification of the major chemical elements by SEM-EDS resulted in a ratio of 2.0–2.2 between Ca and P for both modern and fossil enamel, characteristic of hydroxyapatite. The ICP-MS analyses recovered 27 microelements (in addition to Ca) from superficial enamel samples. For the majority of chemical elements, it was not possible to establish differences between modern and fossil enamel. Nevertheless, an increase in the concentration of Pb, Co, Cd, Ce, Th, As, Cu, Bi, Ag, and Tl was seen in the fossilized enamel, most likely deposited during permineraliztion, with different magnitudes of change. We believe that future studies should be conducted in order to test if larger sampling agrees with our preliminary results and improves the inference power regarding the resistance of fossil enamel through deep time. Furthermore, we expect to recover and identify the organic matrix of dental enamel from the Miocene species studied here (work in progress). Results of this study show the superior preservation of the dental enamel over long time windows, a tissue particularly important for the current interest in the knowledge of ancient environments and peptide recovery, and, as such, for paleontology of the 21st century.

## Figures and Tables

**Figure 1 biology-11-01636-f001:**
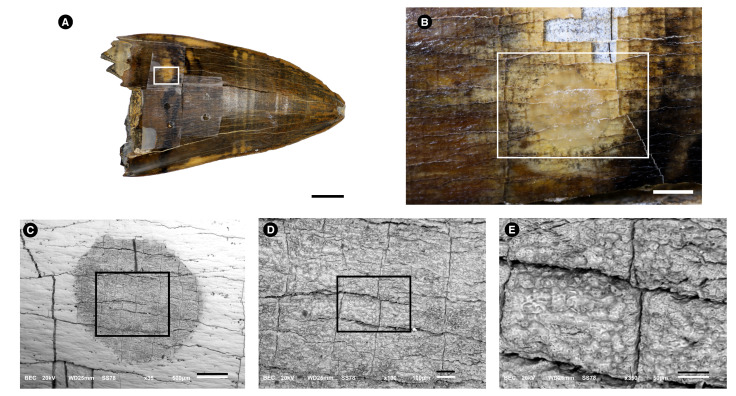
Images of a *Purussaurus* sp.(Crocodylia, Alligatoridae) tooth used to obtain superficial enamel acid etch samples. Images were made with photograph cameras (**A**,**B**) and with a Scanning Electron Microscope using Back-scattered-electrons (BSE) mode (**C**–**E**). Rectangles are always amplified in the next figure of the series. (**A**). *Purussaurus* sp. (Crocodylia, Alligatoridae) tooth picture taken without magnification lens. The white arrow indicates enamel cracks. Bar = 1 cm. (**B**). Details of the white rectangle depicted in (**A**). Bar = 1 mm. In the lower area of the figure, some “squares” have fallen, showing the dentine below the enamel. (**C**). Amplification of the white rectangle shown in (**B**). The circular area (gray) was etched with the acid. Bar = 500 μm. (**D**). Amplification of the rectangle shown in (**C**). Bar = 100 μm. (**E**). Amplification of the rectangle shown in (**D**). Bar = 50 μm.

**Figure 2 biology-11-01636-f002:**
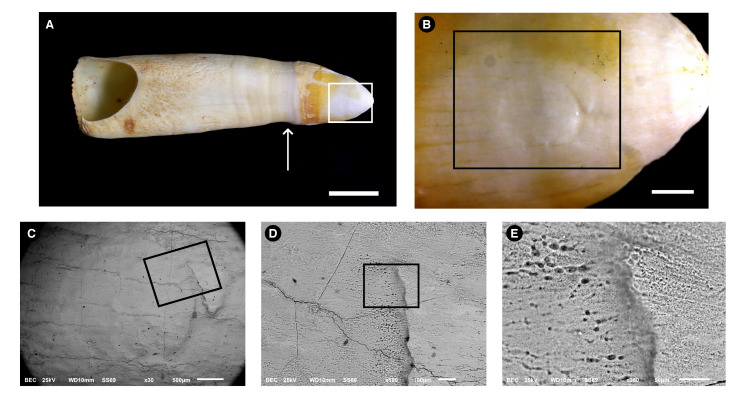
Images of a *Melanosuchus niger* (Crocodylia, Alligatoridae) tooth used to obtain superficial enamel acid etch samples. Images were made with photograph cameras (**A**,**B**) and with a Scanning.

**Figure 3 biology-11-01636-f003:**
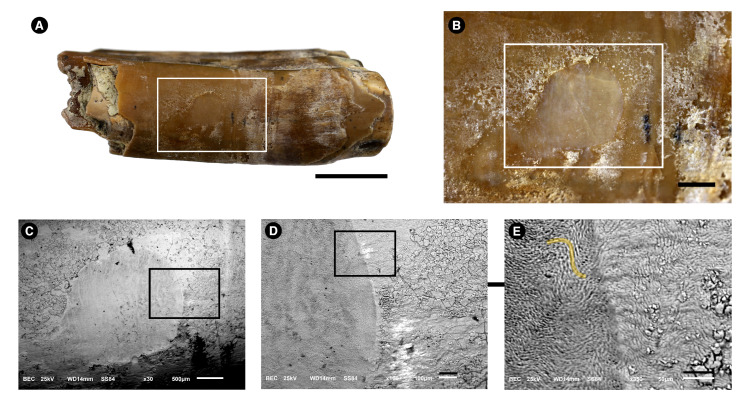
Images of a *Neoepiblema* sp. (Rodentia, Caviomorpha) lower incisor used to obtain superficial enamel acid etch samples. Images were made with photograph cameras (**A**,**B**) and with a Scanning Electron Microscope using Back-scattered-electrons (BSE) mode (**C**–**E**). Rectangles are always amplified in the next figure of the series. (**A**), tooth fragment picture showing the superficial enamel (brown aspect) that has some white deposits and is broken on the left side of the tooth, where the underlying dentine is apparent. Bar = 0.5 cm. (**B**). Details of the white rectangle depicted in (**A**). Bar = 1 mm. (**C**), amplification of the white rectangle shown in **B**. The circular area (grey) was etched with the acid. Bar = 500 μm. (**D**), amplification of the black rectangle shown in (**C**). Bar = 100 μm. (**E**). Amplification of the black rectangle shown in (**D**). In the etched area (left), the enamel’s rods/prisms (yellow line) are apparent. Bar = 50 μm.

**Figure 4 biology-11-01636-f004:**
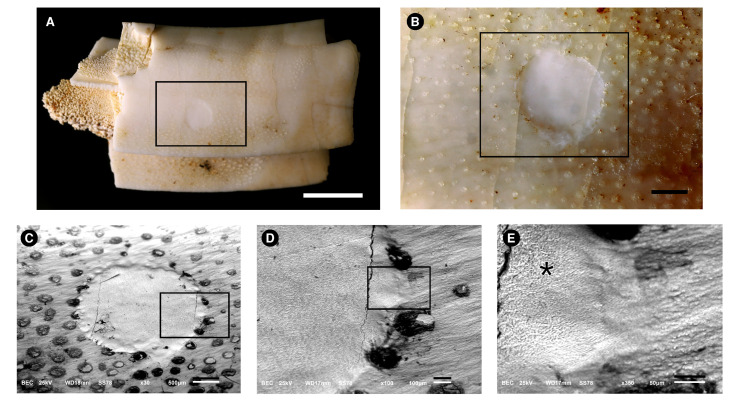
Images of a *Hydrochoerus hydrochaeris* (Rodentia, Caviomorpha) tooth used to obtain superficial enamel acid etch samples. Images were made with photograph cameras (**A**,**B**) and with a Scanning Electron Microscope using Back-scattered-electrons (BSE) mode (**C**–**E**). Rectangles are always amplified in the next figure of the series. (**A**), tooth picture taken with magnification lens showing the circular etched area inside the black rectangle. Bar = 0.5 cm. (**B**), details of the rectangle depicted in (**A**). Circular transparent spots with a diameter of ≈200 μm can be observed. Bar = 1 mm. (**C**), amplification of the rectangle shown in (**B**). The circular area was etched with acid. The circular spots are superficial, having disappeared in the etched area, and have a composition with less dense elements (suggesting an organic composition) than the Ca and P that compose the hydroxyapatite of the enamel. Bar = 500 μm. (**D**), amplification of the rectangle shown in (**C**). Bar = 100 μm. (**E**)., amplification of the rectangle shown in (**D**). Prismatic enamel is seen on the left (asterisk) in the etched area. Bar = 50 μm.

**Figure 5 biology-11-01636-f005:**
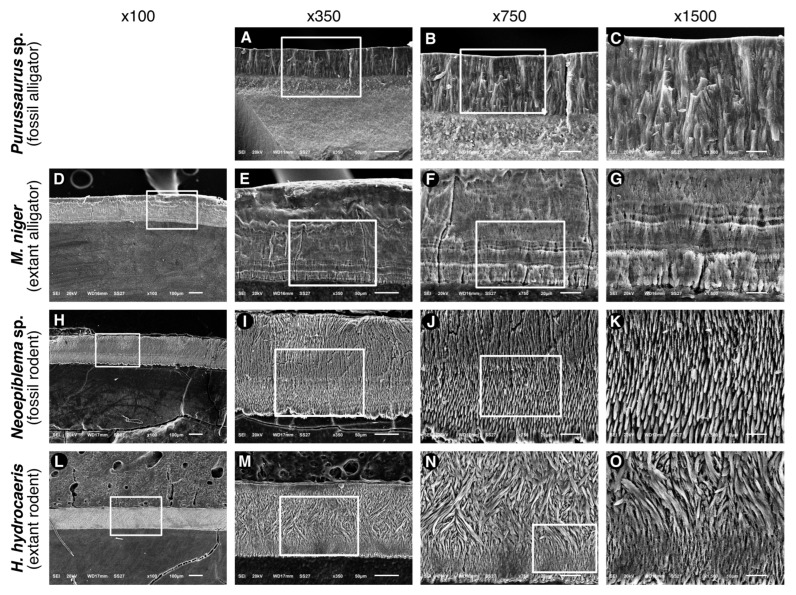
Images obtained by SEM analysis of enamel cross-sections from the four (4) specimens *Purussaurus* sp., *Melanosuchus niger*, *Neoepiblema* sp., and *Hydrochoerus hydrochaeris*. (**A**–**C**), the enamel of *Purussaurus* sp. is formed by rod-like structures that run roughly perpendicular to the enamel surface and dentine-enamel junction (DEJ), having a width of approximately 5 μm. (**D**–**G**), *Melanosuchus niger* has a typical aprismatic structure, showing growth lines that are likely to be the equivalent to the mammalian Retzius lines. These lines were localized near the DEJ and are likely to represent periods of physiological stress during enamel synthesis. (**H**–**K**), the enamel of *Neoepiblema* sp. is formed by prisms that are roughly parallel and perpendicular to the DEJ and enamel surface. The DEJ exhibits an irregular surface. (**L**–**O**), the enamel of *Hydrochoerus hydrochaeris* shows 2 distinct patterns of prims arrangement. The prisms run nearly parallel from the DEJ and extend approximately 25–30 μm near the DEJ. A thin layer of parallel prisms is also observed near the enamel surface. In most parts of enamel, the prisms exhibit a pattern of intense decussation.

**Figure 6 biology-11-01636-f006:**
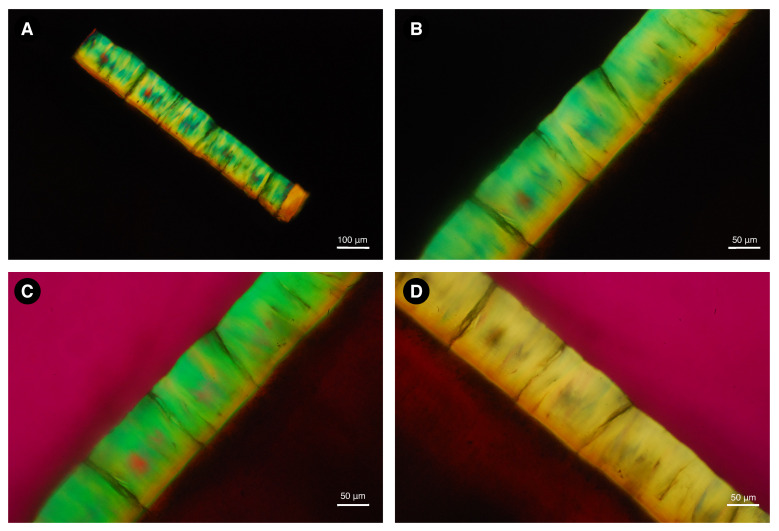
Image of a ground section of dental enamel sample from *Purussaurus* sp. taken under polarizing microscopy with water immersion. Without an interference filter, dental enamel shows high interference colors (**A**,**B**), indicating a relatively high ground section thickness. With the Red I filter, the addition interference color is shown at the −45° diagonal position (**C**), and the subtraction interference color is shown at the +45° diagonal position (**D**), indicating negative birefringence.

**Figure 7 biology-11-01636-f007:**
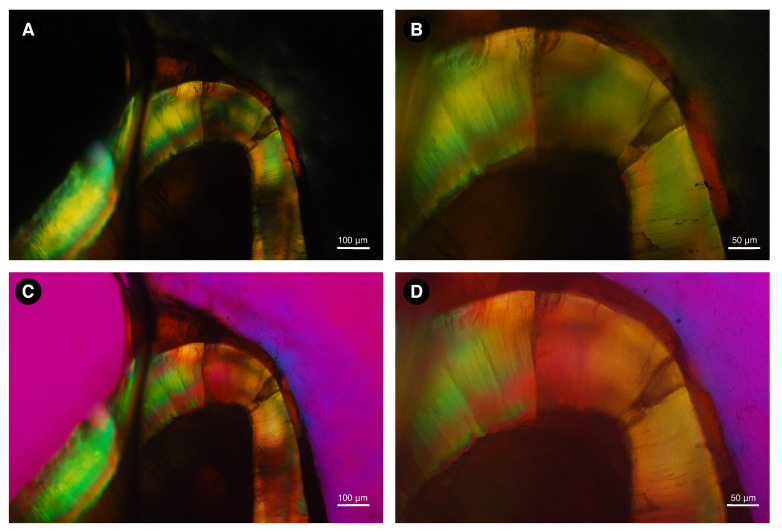
Image of a ground section of dental enamel sample from *Neoepiblema* sp. taken under polarizing microscopy with water immersion. Without an interference filter, dental enamel shows high interference colors (**A**,**B**), indicating a relatively high ground section thickness. With the Red I filter, the addition interference color is shown at the −45° diagonal position (**C**), and the subtraction interference color is shown at the +45° diagonal position (**D**), indicating negative birefringence.

**Figure 8 biology-11-01636-f008:**
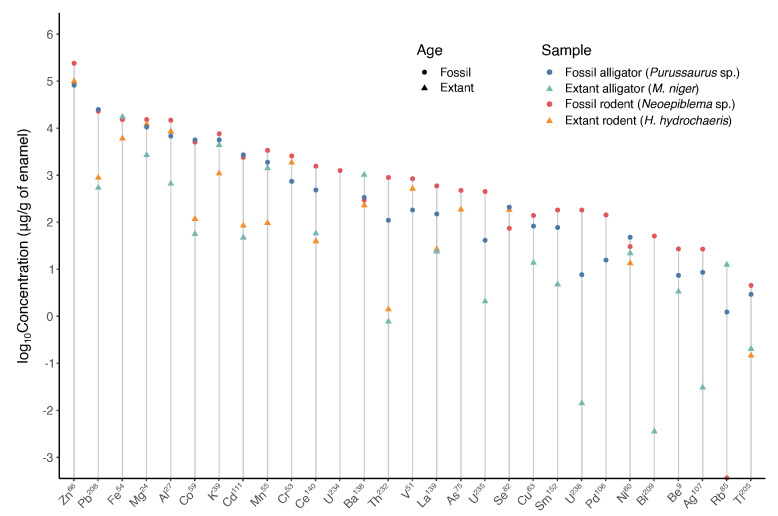
Log${}_{10}$ concentration (μg/g of enamel) of 29 microelements detected in the enamel samples. The fossil samples are identified by a circular icon and the modern samples by a triangular icon. Light and dark blue are used for Crocodylia species, while red and orange are used for Rodentia species.

**Figure 9 biology-11-01636-f009:**
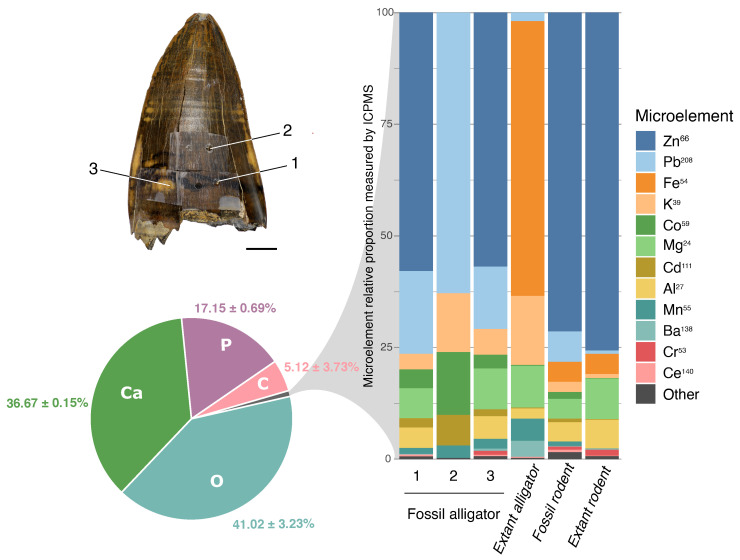
Pie chart displaying the mean (±SD) of the major chemical macroelements (Ca, O, P, and C) found in the enamel (weight percent, wt%) of all samples analyzed by SEM-EDS in this study. On the right, the bar chart shows the proportion of microelements lower than 1% found in each sample (and determined by ICP-MS). In the bars, each element appears as a relative percentage based on the more abundant microelements determined by ICP-MS, except calcium. The picture on the upper left shows the three different enamel regions analyzed in the fossilized alligator (*Purussaurus* sp.) enamel: 1. The darkest enamel area; 2. The predominant brown color enamel; and 3. The yellow enamel area.

**Table 1 biology-11-01636-t001:** Mean ± SD (n = 4–5 measurements in each sample) of macroelement composition (weight%) determined in the internal enamel by SEM-EDS. The first two rows are Crocodylia samples (*Purussaurus* sp., and *M. niger*), and the last two are Rodentia samples (*Neoepiblema* sp. and *H. hydrochaeris*). The Ca/P ratio corresponds to the expected for hydroxyapatite composition. Ca: calcium, P: phosphorus, Ca/P: ratio of calcium per phosphorus.

Sample	Ca (wt%)	P (wt%)	Ratio Ca/P
Fossil alligator	36.8 ± 24.4	16.5 ± 0.2	2.2 ± 0.03
Extant alligator	33.3 ± 0.4	16.6 ± 0.1	2.0 ± 0.03
Fossil rodent	38.6 ± 0.3	17.8 ± 0.1	2.2 ± 0.02
Extant rodent	36.8 ± 1.5	17.7 ± 1.3	2.1 ± 0.12

**Table 2 biology-11-01636-t002:** Enamel components (volume and weight percentages) measured in samples from *Purussaurus* sp. (fossilized alligator), *Melanosuchus niger* (extant alligator), *Neoepiblema* sp. (fossilized rodent), and *Hydrochoerus hydrochaeris* (extant rodent). The "water" component corresponds to adsorbed water during sample preparation.

	Distance from Enamel Surface (μm)	Mineral		Organic ^1^		Water	
		vol (%)	wt (%)	vol (%)	wt (%)	vol (%)	wt (%)
	15	85.3	93.1	8.2	4.5	6.6	2.4
	35	83.8	92.3	9.8	5.4	6.4	2.4
	55	83.3	92.0	10.3	5.7	6.3	2.3
	75	89.2	95.3	4.6	2.5	6.3	2.2
fossil alligator	95	87.7	94.5	5.8	3.1	6.5	2.3
	**Mean**	85.3	93.1	8.2	4.5	6.4	2.3
	**SD**	2.5	1.5	2.5	1.4	0.1	0.1
	15	47.7	65.0	49.0	33.5	3.3	1.5
	55	62.9	77.7	33.9	21.0	3.3	1.4
	95	59.5	75.3	35.4	22.5	5.2	2.2
	135	59.8	75.2	37.6	23.7	2.7	1.1
extant alligator	175	59.2	74.5	39.7	25.1	1.2	0.5
	**Mean**	59.5	75.2	37.6	23.7	3.3	1.4
	**SD**	5.8	4.9	6.0	4.9	1.4	0.6
	40	42.8	61.3	47.1	33.8	10.1	4.9
	70	53.2	70.8	37.3	24.9	9.5	4.2
	100	52.0	69.7	38.8	26.1	9.2	4.1
	130	52.3	69.9	39.7	26.6	8.0	3.6
fossil rodent	160	52.3	69.9	39.2	26.3	8.5	3.8
	**Mean**	52.3	69.9	39.2	26.3	9.2	4.1
	**SD**	4.3	3.9	3.8	3.6	0.8	0.5
	20	71.7	85.4	16.9	10.1	11.5	4.6
	45	69.9	84.2	18.5	11.2	11.6	4.7
	70	69.4	83.9	18.4	11.2	12.3	5.0
	95	68.4	83.2	19.2	11.7	12.5	5.1
extant rodent	120	68.1	82.9	20.2	12.4	11.7	4.8
	**Mean**	69.4	83.9	18.5	11.2	11.7	4.8
	**SD**	1.4	1.0	1.2	0.8	0.4	0.2

^1^ Density estimated based on the density of glutamic acid (1.5 g/cm^3^).

## Data Availability

Not applicable.

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
