# Peer review of "Structure and Chemical Composition of ca. 10-Million-Year-Old (Late Miocene of Western Amazon) and Present-Day Teeth of Related Species"

_biology, 2022, doi:10.3390/biology11111636_

Round 1
Reviewer 1 Report
This manuscript reports an interesting comparison of histology and chemical composition across four teeth, two are fossils and two are extant. The authors are correct that, at least within paleontology and paleobiology, this approach is novel. These histological features and chemical signals are not usually treated as traits for comparison in and of themselves, but are usually used as tools for inferring environmental or dietary signal.
While the authors do often refer to this project as exploratory, they do not make it clear what was the motivation for the study. I recommend that a sentence be added to the beginning of the abstract that explains why this comparison should be done. Also, I think this broader context ought to be emphasized in the discussion more, rather than the very detailed discussion currently presented.
The authors communicate clearly that this study was done on an incredibly small sample size. However, in the discussion they dedicate a lot of text to exploring the differences in chemical composition between the specimens. I recommend de-emphasizing this, as sample sizes of 1 do not justify conclusions of difference. Instead, the emphasis should be on what is observed to be similar, i.e., the major chemical constituents (as is done in the abstract). The differences in the other 27 chemicals provide justification for additional study with larger sample sizes, but not much more.
A few more detailed suggestions:
For readers who are not so familiar with the genus and species names, the manuscript gets a little confusing, especially since the fossil and extant taxa in the the rodent and crocodylia taxa have different genus names. Can you simplify this a bit to help the reader more easily follow along? For example, you could refer to the fossil versus extant rodent, rather than using abbreviations of the Linnaean name.
It would be really nice if the orientations of the teeth in figures 1 through 4 were the same, such that the apex and cervix of the crowns were oriented the same.
The labels on Figure 5, on the left side of the figure and also the panel labels, are very hard to see because the font size is so small.
I don't really follow the analysis presented in figure 6, or what these figures are showing. Can you please expand the explanation of this analysis for readers like me who are unfamiliar with it? Also, is there a reason that the images in Figure 6 are all oriented differently?
There is a typographic mistake on page 14, lines 382 to 383 (sentence ends prematurely).
Reviewer 2 Report
This is an interesting study. I do have a few minor issues for the authors to consider.
With regard to the mineralization of the samples, the authors do not seem to consider in detail that many of the elements present in the samples or differences between the modern and fossil specimens simply reflections differences in the permineralization of the fossil. For example the higher mineral content of the fossil crocodylian compared to the rodent may reflect that they came from different localities and local differences in the mineral content of the sediments from which the fossils were recovered. As they note in their conclusions, larger samples need to be studies and this should include multiple specimens from each locality from which the fossils were recovered.
Page 16, Discussion How can the presence of lead (or any toxic metal) in a fossil be determined to have been incorporated or ingested when the animal was alive, as opposed to have been deposited during the permineralization of the tooth
Page 17 - just an observation but the mineral content and density in the enamel making the tooth of Perussaurus stronger might simply be an adaptation for crushing bone give its larger body size and hence potentially larger prey
In the supplementary material fig 2 is too dark to see any details and a needs a replacement image
I have attached a copy of the manuscript in which I have made a number of suggestions regarding the English. This should not negatively reflect on the research itself but are offered in the hope that the suggested alternative wording will help the authors convey their results and ideas a little more clearly.
I found the study very interesting and feel it should be relatively easy to address the edits and comments provided.
